# Fabrication of LuAG:Ce^3+^ Ceramic Phosphors Prepared with Nanophosphors Synthesized by a Sol-Gel-Combustion Method

**DOI:** 10.3390/mi13112017

**Published:** 2022-11-18

**Authors:** Seok Bin Kwon, Seung Hee Choi, Jung Hyeon Yoo, Seon Yeong Lee, Bo Young Kim, Ho Jung Jeong, Wan Ho Kim, Jae Pil Kim, Bong Kyun Kang, Dae Ho Yoon, Young Hyun Song

**Affiliations:** 1School of Advanced Materials Science and Engineering, SungKyunKwan University, Suwon 16419, Republic of Korea; 2Lighting Materials & Components Research Center, Korea Photonics Technology Institute, Gwangju 61007, Republic of Korea; 3Department of Electronic Materials and Devices Engineering, Soonchunhyang University, Asan City 31538, Republic of Korea; 4Department of Display Materials Engineering, Soonchunhyang University, Asan City 31538, Republic of Korea

**Keywords:** ceramic phosphor, LuAG: Ce^3+^, green phosphor, laser diode, sol-gel-combustion

## Abstract

The aim of this study was to investigate properties of ceramic phosphors fabricated using nano Lu_3_Al_5_O_12_:Ce^3+^ phosphors produced with a sol-gel-combustion method. These nano Lu_3_Al_5_O_12_:Ce^3+^ phosphors had a size of about 200 nm, leading to high density when fabricated as a ceramic phosphor. We manufactured ceramic phosphors through vacuum sintering. Alumina powder was added to improve properties. We mounted the manufactured ceramic phosphor in a high-power laser beam projector and drove it to determine its optical performance. Ceramic phosphor manufactured according to our route will have a significant impact on the laser-driven lighting industry.

## 1. Introduction

Light emitting diodes (LEDs) have fascinated the lighting market for a long time due to their advantages, such as low power consumption, high efficiency, and long reliability [1,2,3,4,5]. However, the constantly developing and evolving technology market demands brighter and more efficient products for better performance. LEDs cannot escape the “efficiency droop” phenomenon that occurs in high-current driving [6,7,8]. For this reason, the current trend in the lighting market is selecting laser diodes as strong candidates for the next generation of light sources. Laser diodes (LDs) are high-power light sources that can achieve excellent performance, where output power increases linearly with rising driving current. They have characteristics such as fast conversion, long throw distance, and high brightness [6,9,10,11]. The combination of laser diodes and converters is driven by a remote type that is unlike the general in-chip LED package, because heat generation is inevitable due to a high output. Types of remote phosphors typically include phosphor in glass, single-crystal phosphor, and ceramic phosphor [12,13,14,15]. Among these types, ceramic phosphors have been studied a great deal in the laser lighting field due to their excellent properties, such as a high light extraction effect, conversion efficiency, thermal conductivity, and thermal shock resistance [16,17].

Garnet-structured oxide phosphors are well known as light converters because of their high thermal properties and their high light efficiency [18]. In addition, their optical properties are not destroyed by high-temperature impact in the process of manufacturing powder into ceramics. A typical example of such a garnet-structured oxide phosphor is a cerium-doped yttrium aluminum garnet phosphor that is used as a material for realizing white light in laser-lighting applications [19]. Lu_3_Al_5_O_12_:Ce^3+^ (LuAG: Ce^3+^) phosphor, which has the same garnet structure in which lutetium is substituted for yttrium, is a well-known type of green phosphor [20,21]. This green LuAG:Ce^3+^ phosphor is manufactured as a ceramic phosphor. It has been widely applied to blue laser-based projection technology. As projection technology is accompanied by high output and long operation time, it is necessary to improve thermal characteristics and light-extraction efficiency of the optical converter used [22,23]. To improve these two properties, the size of the phosphor particle for densification and the addition of functional materials for improving thermal properties can play important roles [24,25].

In this study, nano-sized phosphors were prepared using the sol-gel-combustion method to improve properties of LuAG:Ce^3+^ ceramic phosphor. The sol-gel-combustion synthesis method has the advantage that it can not only manufacture the phosphor in nanosize, but also enable mass production. Furthermore, we prepared a ceramic phosphor by adding alumina to improve thermal properties. It exhibited an excellent value of 2475 lumens with electroluminescence intensity improved by 6.5%, compared with a commercial sample.

## 2. Materials and Methods

Nano LuAG:Ce^3+^ phosphor was prepared via the sol-gel-combustion method. First, 20 g of lutetium oxide was completely dissolved in 200 mL of 70% nitric acid. The remaining raw materials, such as citric acid, aluminum nitrate, cerium nitrate, and propylene glycol, were also dissolved in nitric acid. The completely dissolved solution was adjusted to pH 6.0 with aqueous ammonia. The prepared solution was evaporated while boiling on a hot plate and heat treated at 300 °C to prepare ash. Finally, the ash was put in a mortar, ground, and pulverized. It was then placed in an alumina crucible and calcined at 1400 °C to prepare a nano LuAG:Ce^3+^ phosphor. The alumina (Al_2_O_3_) used in the manufacture of ceramics was purchased from Sigma-Aldrich. The synthesis method is described in Figure 1.

Ceramic phosphor was manufactured through conventional pellet-forming and vacuum-sintering processes. Nano LuAG:Ce^3+^ phosphor and alumina powder were mixed with acetone in a mortar at a 1:1 ratio. The mixed powder was filled in a mold to make pellets using a uniaxial pressure. Pellets were packed in a silicone wrapper and isotropically pressed under 2000 bar with a cold isostatic press. Densified pellets were put into a BN crucible and sintered at 1800 °C in a vacuum atmosphere. After grinding and dicing the sintered body, it was finally oxidized in an atmospheric atmosphere at 1400 °C to remove carburization on the surface and inside. Commercial ceramic phosphors used in laser beam projectors were used as a comparison group.

Electroluminescence spectra were measured by double integrating spheres (PSI Co., Ltd./Korea) under blue laser at 450 nm. The surface morphology of samples was measured by field emission scanning electron microscopy (FE-SEM, JEOL, JSM-7600F). XRD patterns of the LuAG:Ce^3+^ powder and ceramic phosphor were performed over the range of 20° ≤ 2θ ≤ 80° using a diffractometer (XRD, D8 Advance, Bruker) operated at 40 kV and 40 mA with the CuKα target. A ceramic phosphor was attached to the copper plate for laser projection. Illuminance (lx) was measured by attaching a ceramic phosphor to a handmade beam projector.

## 3. Results

The SEM images of nano LuAG:Ce^3+^ phosphor particles prepared by the sol-gel-combustion method are shown in Figure 2. The SEM results confirmed that a single particle included in the aggregated particle was about 200 nm in size. The combination of the instantaneous high temperature during combustion and the release of large amounts of volatiles from the mixture was prone to particle aggregation. However, because the final application of the material was a ceramic aimed at high densification, aggregation between nanoparticles was not disturbed at all.

The SEM and EDS results of the top view of the ceramic phosphor shown in Figure 3a–c were performed for elemental analysis of Al_2_O_3_ and LuAG:Ce^3+^. In Figure 3a, clear contrast and interface were observed between Al_2_O_3_ and LuAG:Ce^3+^ phosphor, indicating that further interaction between particles did not occur. The EDS results in Figure 3b,c showed bright and dark particles confirmed to be LuAG:Ce^3+^ and Al_2_O_3_ particles, respectively. The SEM images also suggested that the ceramic phosphor was highly densified. Thus, the intrinsic characteristics of each composition were well maintained after coarsening and densification. Therefore, high-quality ceramic phosphor was successfully fabricated.

The results of the analysis of photoluminescence (PL) and PL excitation of nano LuAG:Ce^3+^ phosphors prepared by the method depicted in Figure 1 are displayed in Figure 4a. The XRD pattern shown in Figure 4b suggested that LuAG:Ce^3+^ ceramic phosphor fabricated by mixing with Al_2_O_3_ powder coexisted without collapsing. The reason that the phases could be maintained without collapsing the grains, even after sintering at a high temperature of 1700 °C, was that the thermal expansion coefficients of LuAG:Ce^3+^ and Al_2_O_3_ particles were similar [26]. Alumina with a hexagonal structure is well known for its high thermal stability and internal light-scattering-inducing properties. Therefore, if alumina is located in the manufactured ceramic phosphor, an improvement in optical properties and robust thermal properties can be expected during laser irradiation. As shown in Figure 4c, the specimen immediately after vacuum sintering was carburized with a blackish color. This carburizing phenomenon is inevitable, because the main component inside the vacuum sintering furnace is carbon. A carburized specimen can be easily removed through oxidation treatment at 1400 °C. It has an excellent color and an improved PL intensity. To confirm the performance of the manufactured ceramic phosphor, we analyzed the EL spectrum with a comparative specimen. The results are shown in Figure 4d. The prepared ceramic phosphor obtained a maximum emission intensity that was 6.5% higher than that of the comparative group. The nano LuAG:Ce^3+^ phosphor fabricated by our combustion synthesis method was proven to be very effective in ceramic applications.

To evaluate the optical performance of the LuAG:Ce^3+^ ceramic phosphor applied to a laser-driven projection system, a prototype was fabricated, as shown in Figure 5a. It was confirmed that the blue light from the laser light source was successfully converted into green light. Subsequently, in order to measure the projection uniformity according to the location, five points were designated in the projected screen and the illuminance was measured. The measured illuminance values for each part are shown in the inset table on the left of Figure 5b. The mean value and the standard error of the mean were 2414 and 18.76 lx, respectively, indicating uniform illumination. The right inset of Figure 5b plots the spectrum of green-converted light by LuAG:Ce^3+^ ceramic phosphor on the CIE chromaticity diagram. The calculated chromaticity coordinate was (0.3218, 0.6389), which was located in the green region. This suggests that LuAG:Ce^3+^ ceramic phosphor is a significant potential candidate as a green color conversion material for the laser-driven projection system.

## 4. Conclusions

We prepared a ceramic phosphor that can be applied to laser-driven projection and lighting using nano LuAG:Ce^3+^ phosphors synthesized by a sol-gel-combustion method. We added alumina as a functional material to improve thermal and optical properties. We built a blue laser-based beam projector and installed the manufactured ceramic phosphor as an optical converter and drove it. Its optical properties were detected with a measuring instrument at a distance of 1.5 m in a dark room. As a result, the average value of illuminance measured at five designated locations was 2414 lx. This is an impressive result for a high-power, high-brightness laser-based projection technology.

## Figures and Tables

**Figure 1 micromachines-13-02017-f001:**
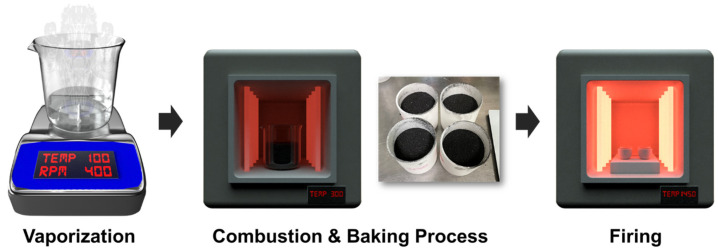
Schematic of the manufacturing process of nano phosphors through sol-gel combustion.

**Figure 2 micromachines-13-02017-f002:**
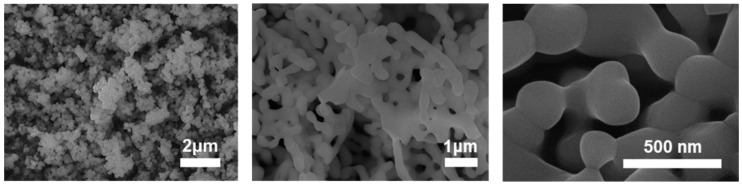
SEM images of LuAG:Ce^3+^ nanophosphor prepared by sol-gel combustion method.

**Figure 3 micromachines-13-02017-f003:**
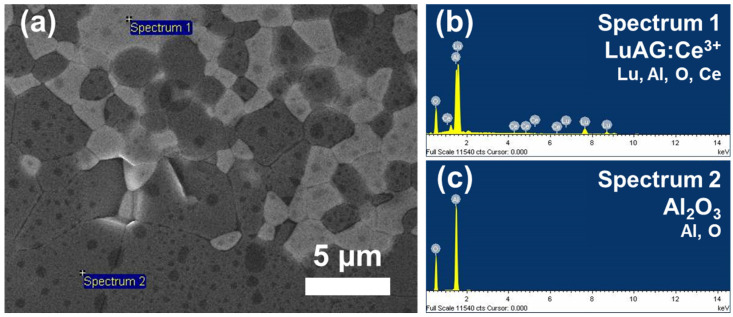
(**a**) Cross-sectional SEM image of the prepared ceramic phosphor; (**b**) and (**c**): point EDS spectra of LuAG:Ce^3+^ and Al_2_O_3_, respectively.

**Figure 4 micromachines-13-02017-f004:**
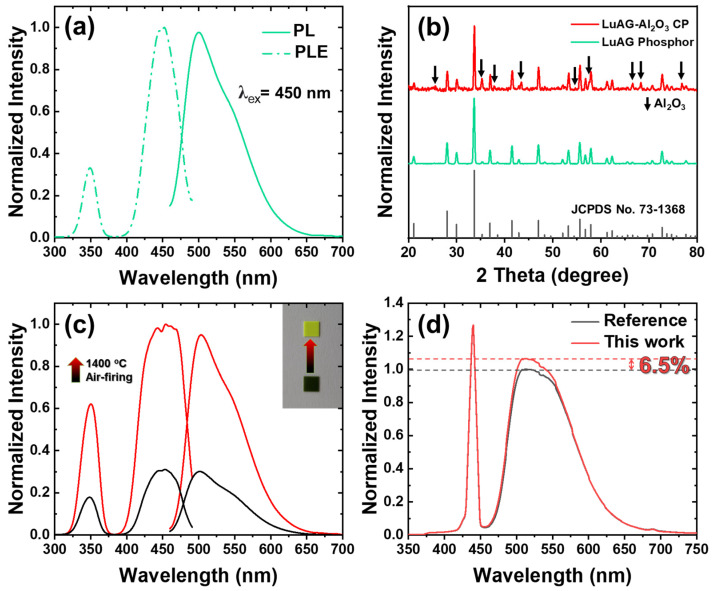
(**a**) PL and PLE of phosphor powder; (**b**) XRD patterns of LuAG:Ce^3+^ phosphor powder and LuAG-Al_2_O_3_ ceramic phosphor; (**c**) PL and PLE before and after carburization removal of specimen; (**d**) EL spectra with comparative specimens.

**Figure 5 micromachines-13-02017-f005:**
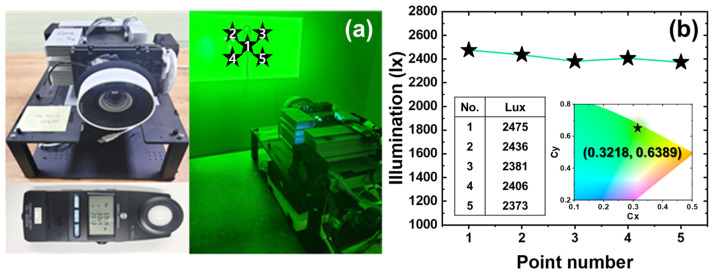
(**a**) Image of a beam projector in operation with the manufactured ceramic phosphor mounted; (**b**) illumination measured at five designated locations at a distance of 1.5 m.

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
