# Peer review of "Fabrication of LuAG:Ce3+ Ceramic Phosphors Prepared with Nanophosphors Synthesized by a Sol-Gel-Combustion Method"

_micromachines, 2022, doi:10.3390/mi13112017_

Round 1

Reviewer 1 Report

The author reported on the research results of applying the manufactured green light-emitting ceramic phosphor to a beam projector. In particular, for novelty, LuAG:Ce3+ phosphor was synthesized by sol-gel combustion. The optical properties of the manufactured ceramic phosphor displayed excellent performance over 2400 lx. However, before recommending for publication, the following issues are suggested to be further addressed.

Reviewer comment #1

As can be seen from the SEM image, the particle size of the manufactured phosphor is shown to be small, but there appears to be a degree of aggregation. Therefore, authors should additionally perform PSA analysis for accurate particle size determination.

Reviewer comment #2

The lx values measured by the author are considered very high characteristics for laser beam applications. However, for readers unfamiliar with this field, it is desirable to provide a table with other references that show how much characteristic values are usually available. 

Reviewer comment #3

Authors should add information about the alumina used for property enhancement to the experimental section.

Reviewer comment #4

The specification information of the beam projector used in Figure 5 should be added. (Example: output power, model, etc.)

Author Response

Response to Reviewer’s comments

Dear Reviewers,

We thank the reviewers for taking a keen interest in our work and their cogent evaluations that help to improve this text as a scientific paper. We have revised the manuscript in line with the editor and reviewers comments and made every attempt to address all the comments as thoroughly as possible.

We look forward to hearing from you.

Sincerely yours,

Response to Reviewer’s comments;

Reviewer #1:

The author reported on the research results of applying the manufactured green light-emitting ceramic phosphor to a beam projector. In particular, for novelty, LuAG:Ce3+ phosphor was synthesized by sol-gel combustion. The optical properties of the manufactured ceramic phosphor displayed excellent performance over 2400 lx. However, before recommending for publication, the following issues are suggested to be further addressed.

Comment #1

As can be seen from the SEM image, the particle size of the manufactured phosphor is shown to be small, but there appears to be a degree of aggregation. Therefore, authors should additionally perform PSA analysis for accurate particle size determination.

Response

We appreciate the Reviewer’s valuable comments. First of all, we would like to apologize. We tried particle size analysis with our PSA instrument but could not get reliable data. This is because the density of the particles is high, so the dispersion in the solvent is not uniform, and it sinks. As you mentioned, it is difficult to define the size of each particle because the particles are agglomerated. So, we expressed the sentence as "the aggregated particle was about 200 nm in size".

Comment #2

The lx values measured by the author are considered very high characteristics for laser beam applications. However, for readers unfamiliar with this field, it is desirable to provide a table with other references that show how much characteristic values are usually available. 

Response

We appreciate the Reviewer’s valuable comments, the illuminance values we analyzed are ambiguous to compare with other literature. For example, in the case of other literature, most of the analysis was done through a laser-based integrating sphere device. However, we measured the amount of light shining on the subject for actual laser projection applications.

Comment #3

Authors should add information about the alumina used for property enhancement to the experimental section.

Response

The information of Al2O3 was added according to the reviewer's comments.

Comment #4

The specification information of the beam projector used in Figure 5 should be added. (Example: output power, model, etc.)

Response

We appreciate the Reviewer’s valuable comments, we used a handmade beam projector equipped with a blue laser. In this regard, it has been corrected to be mentioned in the Experimental section.

Reviewer 2 Report

The article shows the preparation of LuAG-Al2O3, characterization of its properties, and its applications. The article needs to be major revision before it can be accepted

1. The excitation emission wavelength of the sample needs to be marked in Figure 4a

2. In Figure 4b, why the XRD diffraction peak intensity of LuAG-Al2O3 is much higher than that of LuAG phosphor

3. What does the control group in Figure 4d refer to? What is the source of the data? Why the luminescence intensity of the samples prepared in this paper is higher than that of the control group, please explain.

4. How do the samples prepared in this paper perform in laser projection, compared to the control group in Figure 4d? Please make a comparison

5. What is the innovation of the samples prepared in this paper compared with other LuAG-Al2O3 reported?

Author Response

Response to Reviewer’s comments

Dear Reviewers,

We thank the reviewers for taking a keen interest in our work and their cogent evaluations that help to improve this text as a scientific paper. We have revised the manuscript in line with the editor and reviewers comments and made every attempt to address all the comments as thoroughly as possible.

We look forward to hearing from you.

Sincerely yours,

Response to Reviewer’s comments;

Reviewer #2:

The article shows the preparation of LuAG-Al2O3, characterization of its properties, and its applications. The article needs to be major revision before it can be accepted

Comment #1

The excitation emission wavelength of the sample needs to be marked in Figure 4a

Response

We appreciate the Reviewer’s valuable comments, figure 4a was modified.

Comment #2

In Figure 4b, why the XRD diffraction peak intensity of LuAG-Al2O3 is much higher than that of LuAG phosphor?

Response

We appreciate the Reviewer’s valuable comments, the figure was modified as it was determined that it could confuse readers. The two XRD diffraction peaks were separated so that they did not overlap. When sintered with ceramic, the crystallinity is improved, so the peak generated on the diffraction plane may appear stronger, and also it is thought that it may vary depending on the amount of sample (specimen).

Comment #3

What does the control group in Figure 4d refer to? What is the source of the data? Why the luminescence intensity of the samples prepared in this paper is higher than that of the control group, please explain.

Response

We appreciate the Reviewer’s valuable comments, we selected a ceramic plate manufactured by SCHOTT as a control. As can be seen in Fig. 3(a), the phosphor particles that are individually separated are very small. Therefore, we are convinced that the sol-gel combustion method we used induces sintering densification in the manufacture of ceramic plates. It was considered that it could be a sensitive part to mention a specific company name, so it was referred to as a "commercial ceramic phosphors".

Comment #4

How do the samples prepared in this paper perform in laser projection, compared to the control group in Figure 4d? Please make a comparison?

Response

We appreciate the Reviewer’s valuable comments, we attached a ceramic phosphor to the copper plate and applied it to laser projection. Measurements were made under the same conditions as the control group.

Comment #5

What is the innovation of the samples prepared in this paper compared with other LuAG-Al2O3 reported?

Response

We appreciate the Reviewer’s valuable comments, compared with the previously reported literature, we think that the phosphor made by the sol-gel combustion method induces relatively more efficient sintering densification.

Round 2

Reviewer 2 Report

当前修订版是可以接受的